# Cationized Cellulose Materials: Enhancing Surface Adsorption Properties Towards Synthetic and Natural Dyes

**DOI:** 10.3390/polym17010036

**Published:** 2024-12-27

**Authors:** Arvind Negi

**Affiliations:** Faculty of Educational Science, University of Helsinki, 00014 Helsinki, Finland; arvindnegi2301@gmail.com or arvind.negi@helsinki.fi

**Keywords:** cellulose, cationization, fibers, coloration, reactive dyeing, anionic dyes, polyphenols

## Abstract

Cellulose is a homopolymer composed of β-glucose units linked by 1,4-beta linkages in a linear arrangement, providing its structure with intermolecular H-bonding networking and crystallinity. The participation of hydroxy groups in the H-bonding network results in a low-to-average nucleophilicity of cellulose, which is insufficient for executing a nucleophilic reaction. Importantly, as a polyhydroxy biopolymer, cellulose has a high proportion of hydroxy groups in secondary and primary forms, providing it with limited aqueous solubility, highly dependent on its form, size, and other materialistic properties. Therefore, cellulose materials are generally known for their low reactivity and limited aqueous solubility and usually undergo aqueous medium-assisted pretreatment methods. The cationization of cellulose materials is one such example of pretreatment, which introduces a positive charge over its surface, improving its accessibility towards anionic group-containing molecules or application-targeted functionalization. The chemistry of cationization of cellulose has been widely explored, leading to the development of various building blocks for different material-based applications. Specifically, in coloration applications, cationized cellulose materials have been extensively studied, as the dyeing process benefits from the enhanced ionic interactions with anionic groups (such as sulfate, carboxylic groups, or phenolic groups), minimizing/eliminating the need for chemical auxiliaries. This study provides insights into the chemistry of cellulose cationization, which can benefit the material, polymer, textile, and color chemist. This paper deals with the chemistry information of cationization and how it enhances the reactivity of cellulose fibers towards its processing.

## 1. Introduction

Cellulose is the most abundant biopolymer on the planet and can be found in different forms. It can be derived from various natural sources, with wood pulping significantly contributing to the production of commercial-grade cellulose materials [1,2,3]. Cellulose has limited applications due to its physicochemical properties, which are influenced by its unique molecular structural properties. As a homopolymer, cellulose is made up of linear β-glucose units that are linked by 1,4-beta linkages, as shown in Figure 1. The cellulose structure forms intermolecular hydrogen-bonding networks that significantly contribute towards its crystallinity, additionally providing it with water retention capability. Due to high intermolecular as well as intramolecular hydrogen bonding between the hydroxy groups of cellulose structure, cellulose exhibits poor nucleophilicity or reactivity and, therefore, is categorized (in general) as less reactive material or inert material [4,5,6,7,8]. As most of the hydroxy groups are involved in H-bonding, they tend to exhibit low ionic characteristics, limiting their availability for nucleophilic reactions. It is a natural polyhydroxy polymer that is able to show additional H-bonding with water molecules (to a certain extent) and exhibits a limited aqueous solubility with relatively poor organic solvent solubility. However, its solubility profile can be changed depending on its polymorphism, size (micro-to-nano), crystallinity versus amorphous percentage, and other materialistic qualities [9]. These physical characteristics of cellulose not only affect its solubility profile but also other physicochemical properties (for example, its rheological properties). Therefore, depending on the specifics of the cellulose sample, the profile of physical properties can be different. The polyhydroxy nature provides it with a certain degree of water solubility; therefore, pretreatment strategies or processing that is developed to overcome its low reactivity of cellulose materials is often performed in aqueous solutions.

One such pretreatment method is cationization, which introduces positive ionic charges onto the cellulose surface, making it more accessible for ionic interactions. The cationization of cellulose provides various applications in materials science, which are not only limited to antimicrobial finishing [10,11,12] and enhancing reactive dyeing [13,14,15,16,17,18], but many more examples can be seen from different material applications. In particular, the dyeing industry exploits the cationization method as it enhances ionic interactions with anionic dye groups (such as sulfate, carboxylic, or phenolic groups), reducing the dependence of the dyeing process on additional chemical auxiliaries. This study delves into the chemistry underlying the cationization process of cellulose materials. By exploring the mechanisms and reactions involved in dye-fiber chemistry, this paper offers valuable insights that can significantly benefit professionals in various fields. Material scientists can leverage this knowledge to develop advanced cellulose materials with enhanced properties. Polymer chemists can apply these findings to create more efficient and sustainable cellulose processing and functionalization, for example, developing polyvinyl chloride (PVC)-cellulosic nanocrystals or fiber composites [19,20,21,22]. Textile chemists can utilize the information to improve dyeing processes and fabric treatments, leading to better quality and more vibrant textiles. Additionally, color chemists can benefit from understanding how cationized cellulose interacts with different dyes, enabling them to innovate and optimize dye formulations for a wide range of color applications.

## 2. Reactivity of Industrial Cellulose Materials

The cellulose fiber industry encounters numerous obstacles due to the complex chemistries involved in its processing [23,24,25,26,27]. Despite being the most abundant natural polymer, it is not widely accepted in commercial industries because of its inert nature, which is linked to the hydrogen bonding network and crystallinity in its molecular structure [28,29,30]. For example, compared to synthetic fibers, cellulose-based fibers need a higher dye concentration per liter since dye uptake on native cellulose fibers is typically low. To enhance reactivity or improve chemical penetration, cellulose-based materials are often preactivated before any specific functional processing [31,32,33,34]. For instance, commercial reactive dyeing of cellulose fibers usually occurs in an aqueous medium, necessitating the addition of electrolytes (such as alkali metal chlorides or alkali metal sulfates) to increase dye uptake due to the low affinity of reactive dyes for cellulosic fibers. Therefore, cellulose industries are known for their high electrolyte usage in the form of salts and various types of alkali. Cellulose fibers have limited reactivity, necessitating dyes with at least one reactive group to form covalent bonds. The addition of alkali salts like Na_2_CO_3_ or NaOH ensures dye fixation on the fiber by facilitating covalent bonding between the hydroxyl groups of cellulose and the dye molecules [35]. This process involves electrophilic reactions, often targeting the primary OH groups at the C6 position of cellulose.

The limited reactivity of cellulose materials requires a functionalizing agent (dyes) with at least one reactive group to form covalent bonds. For example, the cellulose fiber industry has developed specific dyes, known as reactive dyes, which contain reactive substructures like triazine or vinyl sulphone derivatives. These dyes can be bifunctional or monofunctional, depending on the nature of the reactive groups. For example, if a dye has two different reactive groups (as seen in RR229), it is called a heterobifunctional reactive dye. Conversely, if the reactive groups are the same (as in RR141), it is referred to as a homobifunctional reactive dye, as shown in Figure 2. However, dyes with only one reactive group are known as homofunctional reactive dyes. Since these are common, the term “homofunctional” is not frequently used. Reactive dyes, which have monochlorotriazine, dichlorotriazine, and vinyl sulphone as reactive groups, are commonly used in industrial applications. However, these dyes are prone to hydrolysis in water and alkaline solutions, leading to the loss of unfixed dyes in the dyeing wastewater. Although the inclusion of reactive functionalities like triazine triphenyl derivatives is known for imparting good fastness properties, a certain percentage of these dyes remain unreactive and, ultimately, are discarded in wastewater effluent.

The high electrophilicity of reactive groups, due to the presence of electronegative elements like quaternary ions or halogen substituents attached to the carbon atom, allows them to undergo nucleophilic attack reactions. The propensity or facilitation of such nucleophilic attack on a specific highly electrophilic carbon atom is due to its attachment with a highly electronegative substituent (in the form of an atom, group, or functionality). For example, the chemical structure of a monochlorotriazine group has a chloro-substituent on its electron-deficient triazine ring, which provides a site for nucleophilic attack from hydroxy groups of cellulose and forms non-ionic covalent bonds. Due to the inherent flaws in reactive dyes, which contain highly aqueous subliming sulfate groups, a salting-out/salting-in process is required to improve dye migration towards the cellulose fiber. This necessitates the addition of high amounts of electrolytes, ranging from 30 to 100 g/L, which enhances dye exhaustion and is also known to influence color depth [35]. To enhance reactivity, hydroxyl groups of cellulose need to be exposed to an alkali solution so that the hydroxy group can be deprotonated. Such processes are often achieved with alkaline solutions. This breaks intermolecular hydrogen bonds and deprotonates some OH groups, increasing the availability of free OH groups at the C2, C3, and C6 positions for dye fixation. The nucleophilicity of the C6 hydroxy group, due to its primary alcohol nature, allows it to react with reactive dye groups like monochlorotriazine or vinyl sulphonyl, forming covalent bonds during dyeing. The secondary purpose of adding high amounts of salt in cellulose fiber dyeing is to maintain the cellulose in its ionic form (such as soda cellulose) when sodium chloride is used. The resulting deprotonated glucose units of soda cellulose enhance the negative character (nucleophilicity) of the hydroxy groups, making them available to form covalent bonds with reactive dye molecules (reactive dyes are often highly electrophilic in nature). To an extent, the high electrolyte concentration reduces the water solubility of dyes/anionic molecules, thereby increasing their migration towards cellulose structure (materials such as fibers). However, reactive dyes are sensitive to pH changes in aqueous conditions and can hydrolyze in slightly acidic or alkaline solutions, reducing their effectiveness and causing significant dye waste. This waste can pollute waterways, highlighting the environmental impact of reactive dyeing processes. Currently, the coloration of cellulose materials with reactive and direct dyes is an energy-intensive process that consumes a large volume of water, contributing to water-contaminated solutions. Being anionic in nature, as shown in Figure 2, these dyes tend to have high aqueous solubility, thereby exhibiting a poor migration/affinity towards the cellulose fiber. Additionally, cellulose structure in aqueous solution tends to achieve negative surface charges on its surface, which thereby repel these anionic dyes and discourage their structure adoption (or dyeing) onto the cellulose materials. To overcome this shortcoming, usage of high concentrations of electrolytes like sodium chloride and sodium sulfate are added to the reaction solution (dyebath).

Commercial cellulose materials or fibers are typically dyed using anionic dyes, such as reactive or direct dyes, except acidic dyes. However, this process is often criticized for its high ecological impact, with the textile industry being a major contributor to water pollution. Natural colorants could offer a solution to these environmental issues, but their poor exhaustion rates make conventional dyeing methods unsustainable. Therefore, affordable and competitive solutions from new technological advancements are needed. One alternative is to enhance the reactivity of cellulosic structures (materials) towards anionic compounds (dye) by introducing functionalization (as a pretreatment). For example, cationizing the cellulosic structure (materials such as fibers) enhances its reactivity towards anionic dyes. This would include anionic dyes but mainly limited to reactive dyes and direct dyes. However, natural colorants can also be used with other chemical auxiliaries, such as dispersing them with anionic dispersants, while another approach to use directly after those pretreatment methods that undergo in the alkaline medium as the alkaline medium is known to enhance the natural colorants’ aqueous solubility, especially polyphenolic group containing ones. Herein, the current manuscript paper compiles the chemistry and application of different cationic agents that can functionalize different cellulose materials (for example, viscose, cotton, and other cellulose-containing fibers).

## 3. Cationization of Cellulose Materials or Fibers

Pretreating cellulose material (cellulose fibers) before dyeing can greatly enhance dye-fiber affinity, eliminating the need for salt in the dye bath. This pretreatment resulted in cationizing cellulose material (fibers) with surface positive charges, enhancing their interaction with negatively charged anionic dyes. The cationization of cellulose involves chemical modification of its surface properties where hydroxy groups are tethered with cationic agents, as shown in Figure 3. Historical attempts were also made to improve the dyeability of cellulose material (cotton fibers) by blending with protein materials, similar to those found in natural protein fibers (such as wool and silk), onto cellulose fibers. It has been observed that pretreating cationic polymers with a nucleophilic nature with cotton or similar materials allows salt-free reactive dyeing at neutral pH. Dyeing cationized cotton results in better dye uptake and higher color yields. Additionally, the resulting high surface positive charge over cationized cellulose structure or materials (fibers) leads to stronger dye–fiber interactions, ultimately reducing the need for added electrolytes and minimizing rinsing and after washing. Because of the strong interactions between the dye and fiber, dye exhaustion is significantly improved. These dyes are held more firmly through ionic-ionic interactions near the cationic sites of the materials, resulting in lower dye leaching during laundering.

The work of D M Lewis and coworkers from the Department of Color Chemistry and Dyeing, The University of Leeds, U.K [36,37,38], screened numerous chemicals that can be used to provide persistent cationic charge onto the surface of cellulose fiber. For example, an epoxide containing ammonium salt (glycidyl trimethylammonium chloride), an aziridine derivative which has hydroxy group at the opposite side of quaternary salt in the molecule (*N*, *N*-dimethylazetidinium chloride), exhibits strong nucleophilic attack from the cellulose structure because of three-membered epoxide and four-membered aziridine ring containing the additional electronegative atom (oxygen and nitrogen). Compounds that had an acryl or acyl functionality in their structure (for example, the reactive acryl-containing molecules (N-methylol acrylamide), an acyl-containing group chemical (chloro propionyl chloride), and a polymer that has an acryl group (polyepichlorohydrin acrylamide)) also react with cellulose structure. Interestingly, nicotine-containing structures with thioglycolate as a good leaving group (nicotinyl thioglycolate) under alkaline pad-thermofix conditions (promote reaction with cellulose fiber [36]. The chemical modification of cellulose materials (cellulose fibers) by etherification, esterification, grafting, and crosslinking reactions is generally performed by reaction with functional groups (hydroxy groups) present in the fiber. Indeed, these chemical modifications of cotton are subjects of extensive studies; while looking at the mechanism of these compounds, additional experiments shed light on their behavior and whether they require other chemical auxiliaries in order to achieve moderate-to-high color yield. For example, 1,1-dimethyl-3-hydroxyazetidiniumchloride or *N*, *N*-dimethylazetidinium chloride, abbreviated as DMA-AC, which, when treated with cellulose material (cotton) using the pad-bake method, requires strong alkaline media to achieve efficient color shades, which in turn explains that it requires the presence of alkali to form the covalent bonds. Interestingly, it is noted that it maintained a chemical equilibrium with chemicals (dimethyl amine and *N*, *N*-dimethylamino-2 hydroxy-3-chloropropane), demonstrating that the amino group of *N*, *N*-dimethylamino-2 hydroxy-3-chloropropane undergo nucleophilic ring opening attack on next carbon to the quaternary center of azetidinium ion or γ-chloro group to form oligomeric products.

However, DMA-AC pretreated cotton materials demonstrated salt-free and alkali-free reactive dyeing with improved fixation rates. These findings align with those from Glytac A, an epoxide-containing quaternary salt. This suggests that pretreatment with lower molecular weight functionalizing agents, which modify cellulose materials with tertiary or quaternary centers, enhances reactive dye fixation rates. Numerous studies explain the chemistry behind how certain chemical modifications enhance dyeing, considering dye structure, pH conditions, and salt addition [39]. Specifically, the cationization of cellulose fiber has been extensively studied in relation to anionic dyes. Two types of anionic dyes interact with cellulose fiber surfaces: those with reactive parts and those without. Introducing a cationic surface charge to cellulose enhances interaction with anionic dyes, aiding in dye exhaustion. However, for covalent bonding, the dye must have a reactive part, making reactive dyes particularly beneficial after cationization pretreatment. Direct dyes, lacking reactive groups, may have increased their uptake after cationic pretreatment but cannot form covalent bonds with cellulose fibers, limiting cationization’s scope to reactive dyes or molecules with reactive groups. Some studies note that introducing negatively charged functional groups, like carboxylic groups through carboxymethylation, increases the cellulose’s negative charge, reducing anionic dye uptake. For instance, carboxylic groups on cellulose fibers enhance repulsion with negatively charged dye groups, such as sulfate or carboxylic acid groups, decreasing dye uptake. Conversely, cationic fibers show electrostatic interactions with anionic dyes, including reactive, acidic, and direct dyes, and, to some extent, natural colorants with polyphenolic or glycosidic structures. Cationization enhances ionic interactions with natural colorants, improving dyeing properties for cellulose fibers treated with carboxylic group-containing anthraquinones or flavonoids. However, to achieve cationization, a simple method involves introducing amino groups onto the cellulose structure surface, which are then acidified with certain acids. This process creates persistent cationic sites in the form of quaternary ammonium ions. These ions show high interaction and attraction towards anionic groups in dyes. From a physicochemical perspective, this highlights the significance of ionic interactions, where higher cationization results in greater anionic dye exhaustion. Dyes with more anionic groups will have higher sustainability. Importantly, if these ionic interactions are strong enough, there is little to no need for salt, as the dye is already attracted to the cationized fiber due to the negative-positive interaction. The use of alkali can also be reduced, as some studies report the formation of epoxide rings using alkali, leaving residual alkali in the dye bath, which could explain the lack of additional alkali requirement. However, some authors have demonstrated salt-free and alkali-free reactive dyeing using cationization pretreatment, where they neutralize the fiber to study the impact of alkali-free conditions. Despite the advantages of cationizing cellulose materials (fibers such as viscose, cotton, ramie, etc.) for textile applications to enhance the dyeing of yarns, knits, wovens, and garments, its upscaling is limited.

Color Performance: The color performance of cationic pretreatment samples of cellulose material (fiber or fabric) is subjective. One dye may show better exhaustion properties than a similarly structured dye, affecting the color profile, leveling, and fastness properties.Cost of Processing: Most commercial processes have matured after decades of iterative improvements. The introduction of cationic agents can disrupt production lines, and it would take time for commercial industries to achieve maturity in these processes, hampering their profits. Additionally, if cationization does not significantly enhance targeted dye exhaustion, it will increase overall processing costs.Supply Chain Issues: Adding complexity to the process, such as pretreatment, can be seen as an additional step, often discouraging dyers and textile chemists who need to adjust dyeing protocols for specific shades or color strengths.Scalability: Most studies are limited to lab-scale or pilot-scale, making mass usage of cationization pretreatment challenging. It requires pre-exercise or training of personnel to execute the procedure efficiently. Additionally, cationizing agents often produce epoxide in situ, affecting the thermodynamics of the process and thus requiring experienced personnel.Occupational Risk Hazards: These chemicals are known to cause mild symptoms but not fatal ones. However, studies are typically performed under lab conditions with lower volumes. When used in high volumes on an industrial scale, occupational risk hazards need to be assessed.

The processing of commercial cellulose materials, particularly in the dyeing industry, has a significant ecological footprint. However, cationization of cellulose-based materials (fabrics or fibers) offers a potential solution by reducing the volume of dyes and other chemical auxiliaries required in conventional dyeing processes. Ongoing research and development are focused on refining cationization techniques to achieve salt-free, alkali-free, and waterless dyeing. Additionally, efforts are being made to develop cationizing agents within the reactive dye structure, which could eliminate the need for the pretreatment of cellulose materials (fibers and fabrics) on an industrial scale.

In a study investigating the preference for amination in cellulose dyeing, 2-chloro-2-dimethylaminoethyl hydrochloride (DMAE) was used on cellulose material (cotton) [40]. During the functionalization of the cotton fabric, it was expected that the cellulose surface structure would change to include amino functionalities, thereby enhancing the fixation rates of reactive dyes. The study included three samples: one unmodified (native cotton fabric) and two modified with DMAE. The modified samples attained nitrogen contents of 0.386% and 0.795%, as determined by the Kjeldahl method. According to the authors, the amino groups functionalizing the cellulose surface are more nucleophilic than the aliphatic hydroxy groups of cellulose, making them more available to form covalent bonds and thus improving light fastness. Observations indicated that increasing DMAE concentration caused fiber swelling, which facilitated dye movement within the fibers. This swelling increased the porous nature of the fibers, enhancing the diffusion of reactive dye within the cellulose-based fabric and resulting in improved color strength and light fastness [40,41].

### 3.1. Reactive or Synthetic Cationizing Agents for Cellulose

#### Functionalizing Agents: Epoxides, Acryl Derivatives

Various cationic pretreatments for cellulose materials are available, especially those that possess quaternary ammonium terminals, for example, glycidyltrimethylammonium chloride (Glytac A) and similar other epichlorohydrin derivatives. However, these compounds often exhibit poor physical characteristics (such as corrosive, awful odor) and suffer from low substantivity and stability (against pH and heat), especially when they are processed during their commercial application, therefore that limits their widespread use [42]. To overcome these shortcomings, recently, there was some development in polymeric derivatives of Glytac derivatives to process them for cationizing the cellulose materials (fibers).

Interestingly, a synthesis of a quaternary compound that has acrylamide residue (1-acrylamido-2-hydroxy-3-trimethylammoniumpropane chloride (AAHTAPC)), using α-aminoalkyl-ω-quaternaryamine and acryloyl chloride as reactants was developed. AAHTAPC pretreated cellulose fiber (in this case, cotton fabrics), using a pad-bake process under alkaline conditions, undergoes salt-alkali-free dyeing with reactive dyes. By comparison, an improvement in the exhaustion of the reactive dyes onto the cellulose fibers was observed, with improved K/S values (color strength) when compared to the AAHTAPC untreated cellulose fibers. The resulting color shades onto the cellulose fiber were found with reasonable even leveling and brightness, along with reasonable washing fastness and lightfastness [43].

Seong et al. from Seoul National University, Korea (Department of Fiber and Polymer Science) [44], synthesize five cationizing agents using trialkyl amines with epichlorohydrin for cellulosic fibers, as shown in Figure 4. They used equimolar (1.0 equivalents) of trialkyl amines and reacted with epichlorohydrin, which opens the epoxide ring and yields quaternary salts. The resulting quaternary salts were not purified and were applied onto cellulose fibers at a liquor ratio of 50:l with a 0.2% (ow bath) penetrating agent (Triton X-100). Later, the cationized cellulose fibers were washed with tap water and treated with 2% acetic acid solution for 5 min at 40 °C to neutralize them, followed by rewashing and drying at room temperature. Subsequently, the dyeing using different concentrations (ranging from 1 to 4% owf) of C.I. Acid Red 127 was performed. Despite the low degree of cationization, the treated fibers dyed with acid dyes exhibited excellent antimicrobial activity [44]. The method used for antimicrobial testing was Shake Flask CTM 0923, using gram-positive bacteria (*Staphylococcus aureus*).

Cotton can be pretreated with Sintegal V7conc (Chemapol), a commercial cationic agent that is a quaternized polyglycol ether of fatty amine [45]. Dyeing cationized cellulose materials (cotton fabrics) with anionic reactive dyes, even in salt-free conditions, results in increased color intensity with higher concentrations of the cationic agent. This is primarily due to the enhanced ionic attraction between the cationic groups of Sintegal V7conc and the anionic groups of the reactive dyes, ultimately improving dye exhaustion. Notably, cationized cellulose fabrics have shown greater color uniformity and standard fastness levels compared to untreated fabrics. This indicates that the cationic groups of the surfactant play a crucial role in reactive dye absorption, with dye fixation occurring through covalent bonding with the cellulose hydroxyl groups, which can be understood as similar to the conventional dyeing processes.

### 3.2. CHPTAC as Cationizing Agent

One of the key synthetic cationizing cellulose fiber agents that is used is 3-chloro-2-hydroxypropyl trimethylammonium chloride. The 3-Chloro-2-hydroxypropyl)-trimethylammonium chloride, abbreviated as CHPTAC, is the most common cationization agent used for cellulose material. A common method using a CHPTAC agent for cellulose materials also utilizes a hydroxide-containing alkali, which in most cases is sodium hydroxide. During in situ reaction, CHPTAC, when reacted with sodium hydroxide or other alkalis, results in the formation of 2,3-epoxypropyltrimethylammonium chloride (EPTAC). The 2,3-epoxypropyltrimethylammonium chloride, abbreviated as EPTAC, is an in-situ epoxide embedded in a three-membered ring and also has a quaternary ammonium center. Subsequent cellulose material (cellulose fibers, for example, cotton, viscose, ramie, etc.), if available in the solution, reacts with the in situ formed EPTAC. The reaction mechanism for driving the reaction is a high electrophilic character of epoxide and nucleophilicity of cellulose materials or fibers.

In contrast, cellulose materials, which are known for low reactivity (low nucleophilicity), can react with epoxide of EPTAC. From the cellulose structure, hydroxy groups proceed with a nucleophilic attack onto the epoxide ring of EPTAC to form a covalent bond. As can be seen in Figure 5, after cellulose covalent functionalization of EPTAC through its epoxide ring, the resulting cellulose possesses a positive charge on its surface, which allows the negative charge molecules to adsorb on it due to ionic-ionic interactions. Therefore, molecules that have an ionic charge in the structure are attracted towards the CHPTAC-treated fiber surface due to its positive charges. For example, dye molecules that have sulfate or carboxylic groups or other negatively charged groups in the structure exhibit better adoption rates with such positively charged materials. From the perspective of cellulose fiber, anionic dyes, including reactive, direct, and acidic dyes, can be absorbed onto the surface. As a non-volatile organic salt used in an aqueous solution, CHPTAC-associated respiratory complications are not known, and therefore, there are no recorded occupational risks related to its inhalation. While the process where CHPTAC is treated with NaOH to form in situ epoxide EPTAC is quite reactive, handling such solution requires safety measurements as some of the reports indicate that such solution can potentially be genotoxic and carcinogenic. Therefore, industrial effluents containing such contaminated solutions possess ecological toxicity.

Therefore, it forms an in situ epoxide which is reactive enough to form a covalent bond from coming nucleophilic attack from hydroxy groups of cellulose; therefore, it permanently cationizes the cellulose fibers, improving their ability to interact with dyes and therefore enhances the dye-exhaustion on the fiber. However, in situ, epoxide formation requires CHPTAC and NaOH. Therefore, it raises the pH above 12.0, which I believe further enhances the fiber nucleophilicity as hydroxy groups of cellulose tend to deprotonate in alkaline pH, as shown in Figure 6. However, to validate this point, a control experiment should be conducted where CHPTAC is added to a dye bath containing cellulose fiber but not NaOH. Secondly, it appears that CHPTAC consumes some of the hydroxy groups from cellulose fibers. Therefore, one could argue about the availability of remaining hydroxy groups after cationization for reactive dyeing. Therefore, in my opinion, this suggests a trade-off where some of the hydroxy groups are indeed preoccupied with the cationizing agent, assisting the dye molecule in migrating toward the fiber due to the resulting positive charge onto the fiber surface. Another argument is about the generation of ionic charge over the cellulose surface, which seems equivalent to salt-alkali-assisted dyeing (conventional dyeing). Therefore, researchers should conduct a comprehensive techno-economic comparison to evaluate the performance of these methods. Furthermore, the number of negatively charged groups on the dye structure needed to achieve maximum exhaustion towards the fibers is another point of discussion. Although some studies compare the number of sulfate groups critical for dyeing performance, the dyes have different molecular weights, leading to inconclusive results, highlighting the need for more research to establish an efficient dyeing chemistry process. Anionic Dyes with sodium sulfonate groups, as commonly found in the structure of direct, acid, and reactive dyes, indeed exhibit enhanced dyeing results with such cationized cellulose materials.

In another study, the in situ reaction of 2,3-epoxypropyltrimethylammonium chloride with cellulose hydroxy groups was hypothesized [46]. The cationized cellulose material (cotton fabric) showed superior reactive dyeing performance compared to untreated samples in terms of color strength, achieving efficient levelness, exhaustion, and fixation rates at 1% owf. Among the reactive dyes used on pre-cationized cotton samples, Cibacron Brilliant Blue BR-P, which has a monochlorotriazine reactive group, performed poorly [46]. This may be due to solubility issues, as the reactive dyes used in this study had reactive groups, either monochlorotriazine, vinyl sulphone, or dichlorotriazine. The aqueous solubility of dyes depends on their ionic groups. Therefore, more anionic groups often exhibit high solubility for that respective dye, while larger structures or more aromatic rings within the dye structure increase its hydrophobicity and bulkiness. Another reason could be related to the commercial purity of the Cibacron Brilliant Blue BR-P. Commercial dyes are often not 100% pure to keep costs reasonable, typically ranging from 10–60% purity, with the remaining gross weight consisting of inorganic ions. These ions aid in aqueous dyeing but also add to the overall weight of the packaging, which is a concerning practice in the dyeing business from the perspective of sustainable dyeing chemistry. The addition of these inorganic ions enhances dye solubility in water but can limit exhaustion or dye migration toward the fabric surface. Textile researchers often added salts (according to salting-in/salting-out protocols) to improve the migration of dye towards the fiber/fabric. However, the addition of salt helps maintain the charge over the fabric surface by keeping hydroxy groups in their ionic form, as can be seen in the reactive dyeing of cellulose fiber. In my opinion, there could be the possibility of impurities present in the commercial dye, which significantly affected its aqueous solubility, resulting in poor dyeability in salt-free reactive dyeing (exhaustion = 9.7%). However, dye performance improved to 30% exhaustion with the addition of salt, likely due to the salting-in/salting-out effect, which made more dye molecules available for exhaustion. These points aim to provide a broad understanding for researchers working in similar fields and are not conclusive statements, highlighting the need for careful investigation to understand the mechanisms of dyes with similar functionalities.

Hashem, Smith, and Hauser have studied the CHPTAC-based pretreatment of cotton in a strong alkaline solution (sodium hydroxide), either the high-temperature exhaustion method or the cold pad-batch method [47]. The positively charged ammonium functionality in the form of (CH3)_3_N^+^ enables the ionic bonds (salt linkages) formation with negatively charged anionic groups, which can be found in various sulfate groups containing anionic dyes or carboxy-containing dyes. Due to the presence of such high ionic-ionic interaction, especially in the case of anionic dyes that have multiple negatively charged groups, they possess salt-free dyeing properties onto cotton fabric. Additionally, the availability of positively charged ammonium groups in the form of (CH3)_3_N^+^ impart cellulose materials (cellulose fiber) with antimicrobial properties.

The extent of cationization of cotton fabric depends on several key factors, such as the concentration of sodium hydroxide, reaction temperature of the solution, reaction time, solvent used, molar ratio, and the application method. Montazer, Malek, and Rahimi cationized cotton using CHPTAC through a pad-batch process [46]. Subsequently, reactive dyes with different reactive groups were used to dye these pre-cationized samples. Importantly, salt-free dyeing on these cationized samples was observed, possibly due to substantially enhanced ionic interactions between the cationic centers over the cotton surface and anionic groups present on dye molecules. The resulting dyeing on cationic samples was found to have reasonable wash and dry crock fastness compared to the untreated samples, while some had shown improvement in lightfastness.

Hashem explores the potential for a one-stage process that combines cationization with CHPTAC and cotton pretreatment [48]. This process includes enzymatic and oxidative desizing, scouring, bleaching with hydrogen peroxide, and combinations of desizing–scouring and desizing–scouring–bleaching. These pretreatment operations, with or without cationization, are performed using three techniques: cold pad-batch, pad-steam, and exhaustion. Although cationization chemicals are compatible with scouring ingredients and, to a lesser extent, with enzymatic desizing formulations, they are incompatible with oxidative desizing or peroxide bleaching ingredients. The efficiency of the one-step scouring and cationization process is optimized through a detailed investigation of process parameters. The preactivated or cationized fabric can undergo salt-free dyeing, exhibiting high color shade evenness, residual wax content, and nitrogen content [48]. For the exhaustion method performed in an aqueous solution, the highest efficiency of scouring and cationization is achieved when 10 g/L of CHPTAC, 5 g/L sodium hydroxide is added with desized cotton fabric, where molar dyeing ratio or dyeing bath ratio of 20:1 is maintained at 80 °C for 90 min. Although maximum scouring and cationization efficiency in cold pad-batch methodology achieved with 100 g/L of CHPTAC and 49.3 g/L of sodium hydroxide, results in a higher fixation percentage (50%) compared to the exhaustion technique (25%), while scouring efficiency is similar in both methods. Similarly, cold pad-batch methodology utilizes salt-free dyeing for the CHPTAC pretreated fabrics, maintaining level dye shades, nitrogen content, and residual wax content.

Ramasamy and Kandasaamy have studied the impact of different parameters of the cationization efficiency of two cationic agents: CHPTAC and a polyamino chlorohydrin quaternary ammonium compound (Cibafix WFF) [49]. They found that cationization increases dye utilization by 30% without affecting colorfastness and significantly reduces environmental pollution. Pretreated cotton or cationized cotton dyeing requires only 40% of the steam and 50% of the time needed for conventional dyeing. As cationized cotton dyeing is salt-free and alkali-free, it showcases a smaller chemical footprint than the conventional method; therefore, the resulting effluent usually has lower water pollution. Importantly, the addition of salts and alkalis, along with their complexation with dyes, significantly increases the total dissolvent solids levels in conventional dyeing. Therefore, applying such strategies would indeed benefit the lowering of the TDS from dyeing industries effluent. Subramanian and colleagues conducted a similar study with Cibafix WFF, confirming these findings [50].

Ma et al. from State Key Laboratory of Fine Chemicals, Dalian University of Technology, Dalian 116023, China [51] accomplish pretreatment and salt-free dyeing of greige knitted cotton fabrics using three reactive dyes (C.I. Reactive Red 195, C. I. Reactive Yellow 145, and C. I. Reactive Blue 19). To achieve cationization, they used glycidyltrimethylammonium chloride, where Figure 7a showed the fixation percentage of all the three reactive dyes when dyed to a cationized cotton sample (96.74% for C.I. Reactive Red 195, 95.29%, for C. I. Reactive Yellow 145, and 89.72% for C. I. Reactive Blue 19), as shown in Figure 7b–g.

### 3.3. Triazine Derivatives

Reports indicate that cellulose materials can be chemically modified with triazine derivatives containing cationic and anionic groups. These derivatives impart either a negative or positive charge onto the cellulose surface, depending on the type of ionic group present on the functionalizing agent. Altering surface properties, in this way, changes the chemical structure and morphology of cellulose materials, thereby affecting their reactivity, swelling, and crystallinity. As a result, the modified cellulose typically exhibits enhanced diffusion rates and dyeing properties. Studies often show improved exhaustion and fixation rates of molecules (dyes) for modified cellulose with triazine derivatives compared to native, unmodified cellulose materials.

#### 3.3.1. Cationic Reactive Intermediates or Derivatives

Cationic reactive intermediates act as bridging compounds that help cationize cellulose, enhancing fiber swelling. Simultaneously, they provide reactive sites that facilitate the formation of covalent bonds with anionic dyes. This approach has a broad scope of application, extending beyond dyeing to include enzyme immobilization and functionalizing cellulose with hydrophilic compounds to make its surface more hydrophilic. However, when the dye attaches to the reactive intermediates tethered to cellulose, it can alter the color properties, potentially causing a bathochromic shift. These intermediates have shown improved washing fastness of modified cellulose materials compared to untreated samples.

Cationic–anionic triazine mixture: Xie et al. studied the impact of cationic and anionic functionalization on the cellulose surface using triazine derivatives [52]. The authors used a mixture of cationic and anionic agents, 2,4,6-tri-[(2-hydroxy-3-trimethyl-ammonium)propyl]-1,3,5-triazine chloride (abbreviated as Tri-HTAC) as the cationic agent, and 2,4-dichloro[(6-sulfanilic acid anhydrous)-1,3,5-triazine] (abbreviated as Bi-CSAT) as the anionic agent. The mixture (Tri-HTAC: Bi-CSAT = 5:1 weight/weight) was applied to cellulose in an alkaline medium. This modified cellulose material was then subjected to reactive dyeing. This approach is quite interesting as the authors strategically placed both cationic and anionic sites on the cellulose surface. However, it appears that there are more cationic sites than anionic sites, as higher exhaustion rates were observed in the treated samples compared to the untreated ones. Consequently, the fixation was found to be 13–24% higher for the treated samples than for the untreated ones. The diffusion studies, which estimate the degree of mass transfer or dye uptake from the dye bath to the fiber, are correlated with the diffusion coefficients. The dye (Reactive Blue BF-RN) was studied at different temperatures (25, 35, 45, and 65 °C), reaching equilibrium after 60 min. Reactive Blue BF-RN is an anthraquinone chromophore-based heterobifunctional dye (vinyl sulphone and monochlorotriazine). The steady-state diffusion and comparative diffusion coefficients increased with rising temperatures and were found to be 2–3 times higher for treated cellulose compared to untreated cellulose [52].

#### 3.3.2. Reactive Dyes with Cationic Groups

One of the concepts that textile, color, and cellulose chemists have considered is incorporating a direct cationizing group into reactive dyes. This addition would increase the ionic nature of the dye, making it more aqueous soluble and reducing the need for multi-step synthesis to prepare reactive dyes with multiple sulfonates or other water-solubilizing groups. Reactive dyes are commonly used for cellulose materials or fibers, so incorporating a cationizing group directly into the reactive dye structure also eliminates the need for cationic pretreatment. Since the pretreatment of fibers adds new supply lines during commercial applications, small to medium-sized industries are often reluctant to adopt such methodologies. An additional charge to the reactive dye structure leads to higher exhaustion, further eliminating the dependence on salt. Contrary to the direct application of dyes (similar to direct dyeing), where direct dyes usually need high concentrations of electrolytes for effective dyeing, fabrics dyed with these dyes often have poor wetfastness. However, pre-treating cellulosic fibers with cationic agents can reduce or eliminate the need for electrolytes and improve the wetfastness of direct dyeing. Pre-treating cellulose materials (fabrics and fibers) with mono- or bifunctional cationic reactive dyes, which exhibit high electrostatic interactions, allows direct dyeing even in salt- and alkali-free conditions. The cationic functionality incorporated into the monochlorotriazine ring is shown in Figure 8. Generally, a higher degree of dye exhaustion, fixation, and improved wetfastness is observed in cellulose materials cationized with such agents compared to untreated samples. Bifunctional cationic dyes treated cellulose materials (fabric or fibers) showed additional dye-fiber interactions, resulting in substantial improvements in exhaustion and fixation rates compared to monofunctional cationic dyes.

In a study, to prepare a cationic functionality-containing reactive dye, 1,4-diamino-5-nitro-anthraquinone (a chromophore) reacted with cyanuric acid (trichlorotriazine, which acts as the reactive part of the molecule, aiding it to form covalent bonds with hydroxyl groups of cellulose), forming a non-cationic (or ionic) dichloro triazine-anthraquinone-based reactive dye [53]. Such dyes are known in the literature but exhibit poor aqueous solvent solubility, making them better suited for solvent-assisted dyeing. To this dichlorotriazine intermediate, a pyridinium salt tethered to an alkyl amino chain was added, where the amino group performed a nucleophilic attack on one of the positions of the dichlorotriazine intermediate, forming a pyridinium cation-containing monochlorotriazine-anthraquinone-based dye. Incorporating the pyridinium cation into this dye is important as it impacts the aqueous solubility, which is necessary for water-based dyeing. One interesting decision by the author was to incorporate this pyridinium cation tethered to an alkyl chain, thereby avoiding any color shifting during the dyeing process. These pyridinium-based dyes exhibit salt-free exhaustion and fixation, with a marked improvement in light fastness compared to similar anionic dyes used for cellulose fibers or fabrics [53]. A similar synthetic approach was used by Zheng et al. from Nanjing University of Technology, China, to form three dyes: D1, D2, and D3, as shown in Figure 8 [54]. Interestingly, the authors selected bifunctional cationization using a pyridinium cation on one terminal and a quaternary ammonium on the other. However, the ammonium terminal side was transformed into an epoxide to make it the reactive part for these dyes (D1, D2, and D3). The author’s decision not to use a monochlorotriazine-based reactive part helps some industries reduce halogen emissions. Halogen-based reactive groups often produce occupational hazards, so this approach could be seen as a step toward implementing sustainable chemistry in dyeing. These dyes were yielded in powder form, with D1 being red, D2 green, and D3 blue. In dyeing experiments, optimal conditions for achieving high exhaustion and fixation rates were found with sodium carbonate (20 g/L) at 60 °C rather than 90 °C, compared to NaOH (10 g/L at two different temperatures, 60 °C and 90 °C) [54]. In conclusion, since the synthesis of most reactive dyes commonly uses 2,4,6-trichloro-1,3,5-triazine as a precursor molecule, which also acts as a reactive group in the final dye structure, organic chemists could utilize other scaffolds similar to 2,4,6-trichloro-1,3,5-triazine as precursors, especially for pre-cationized cellulose materials [55]. For example, developing dyes with symmetric or non-symmetric trisubstituted heterocyclic systems could benefit from pretreated cationic samples if the final dye structure contains a sufficient number of anionic groups (or negative charge over its structure) in the form of carboxylic, phenolic, thiol, or sulfate groups [31,56,57].

### 3.4. Cationic Polymers

#### Silicon-Ammonium Polymer

Cationic agent CA200 is a silicon-ammonium polymer with the structure formula [(CH_3_O)_3_Si(CH_2_)_3_(CH_3_)_2_-N-C_18_H_39_]^+^Cl^−^. In a study, it was used to cationized cotton materials in concentrations ranging from 1% to 5% by weight [58]. The nitrogen content, measured by the Kjeldahl method, was highest for the 5% CA200 cationized sample (0.247), compared to the native cellulose sample (0.023), which is nearly 10.7 times higher. In dyeing experiments, a significant difference in color strength, in terms of K/S values, was observed, with Remazol Red RB performing better than Remazol Blue R. This difference can be attributed to their structures: Remazol Red RB is heterobifunctional, containing both vinyl sulphonyl and monochlorotriazine reactive parts, whereas Remazol Blue R has only a vinyl sulphone reactive part. Another major difference is the presence of sulfate groups; Remazol Red RB has four sulfonic groups, enhancing its aqueous solubility, while Remazol Blue R has only two, limiting its solubility. CA200-pretreated cellulose material (cotton fabric) shows an improved dyeing rate even without salt, compared to conventional dyeing with salt. The dyed fabric exhibits good wet fastness and light fastness properties. This study emphasizes that cationization of cellulose materials (fibers or fabric) improves dye substantivity, resulting in better color yields, while reducing the total dissolved solids (TDS), biochemical oxygen demand (BOD), and chemical oxygen demand (COD) of the effluent [58]. This approach could help reduce water pollution from textile industries. However, the use of silicon as a cross-linker in this process, while aiding in producing the cationic center over cellulose structure, raises concerns due to its reported silicon-associated toxicity [59,60,61], making its bulk usage a subject of discussion.

### 3.5. Non-Reactive Pretreatments

A few polymers have shown a cellulose affinity and can be desorbed during dyeing, affecting dye uptake/exhaustion or causing dye precipitation. Recent studies have highlighted the effectiveness of ammonium cationic centers of synthetic polymers tethering to cellulose materials using electrostatic and hydrophobic interactions. These polymers comprise simpler hydrocarbons, mainly either aliphatic chains or aromatic substituents. Although nonreactive polymers show promising salt-free dyeing of cellulose materials (fabric and fibers), challenges remain in selecting suitable dyes and achieving uniform dyeing outcomes.

Papapetros et al. from the University of Patras, Greece [15], evaluated the intermolecular interactions that govern the physicochemical processes in fabric dyeing using spectroscopic tools. They studied a cationic copolymer composed of vinyl benzyl chloride (VBC) and vinyl benzyl triethylammonium chloride (VBCTEAM) for dye uptake on cationized cellulose fibers using Remazol Brilliant Blue R and Novacron Ruby S-3B dyes shown in Figure 9a,b. Interestingly, Novacron Ruby S-3B resembles Reactive Red 229, a heterobifunctional reactive dye with monochlorotriazine and vinyl sulfone reactive groups. Remazol Brilliant Blue R is also known as Reactive Blue 19. The authors used ATR-FTIR, UV-Vis, fluorescence, and XPS spectroscopy to study the dye-fabric interactions, highlighting the significance of sulfonate/sulfate groups. The stoichiometric charge ratios of dye-polymer precipitates indicated that the anionic groups of the reactive dyes interact with the quaternary ammonium groups of VBCTEAM units through ionic interactions. This interaction leads to higher dye exhaustion as the number of anionic groups in the dyes increases, thereby enhancing dye uptake [15]. Fang et al. synthesized a cationic polymer, poly[Styrene-Butyl acrylate-(P-vinylbenzyl trimethyl ammonium chloride)], abbreviated as (P(St-BA-VBT), as shown in Figure 9c [62]. The nano desizing of P(St-BA-VBT), which has quaternary ammonium cationic sites, is used to treat the cellulose material (cotton) by the pad-dry process. Subsequently, a reactive pad-steam dyeing procedure was used to dye the P(St-BA-VBT)-pretreated cellulose samples, where optimal conditions were found for polymer (P(St-BA-VBT) = 4 g/L), Na_2_CO_3_ (25 g/L) with a time of steaming (180 s) with C.I. Reactive Blue 222 (25 g/L). Comparatively, significant color yields and fixation were observed for P(St-BA-VBT)-treated cellulose (39.4%) material than untreated one (14.3%), with a reduction of usage of salt and steaming time. Scanning electron microscopy exhibits a random distribution of P(St-BA-VBT) nanospheres on the cellulosic material rather than a continuous film [62].

Cationic polymer with nucleophilic groups: Blackburn et al. from the University of Leeds, UK, studied two cationic polymers (PT1 and PT2) with molecular weights ranging from 10,000 to 30,000 g/mol, as shown in Figure 10a,b [63]. PT1 is a polymer composed of two monomer units: diallyldimethyl ammonium chloride (a cationic monomer with a quaternary center, represented by *m*) and 3-aminopropene (a nucleophilic monomer unit, represented by *n*), present in a ratio between 1:1 and 1:5. PT2 is a polymer of 4-vinylpyridine (a cationic monomer with a quaternary center, represented by *m*) and 1-amino-2-chloroethane (a nucleophilic monomer unit, represented by *n*) in a ratio between 1:3 and 3:1. The high substantivity of these polymers (PT1 and PT2) towards cellulose materials (fabric or fiber) is driven by electrostatic interactions, intramolecular/intermolecular hydrogen bonding, and van der Waals forces. At an application pH of 6–7, functionalities with relatively low pKa values ionize. For example, at this pH, negatively charged carboxylic acid groups of the substrate exhibit electrostatic interactions with the positively charged quaternized centers of the polymer, with additional hydrogen bonding and dispersion forces enhancing the attraction between the polymers and the fibers as shown in Figure 10c,d [63]. Another contributing factor is the presence of nucleophiles in the cationic polymer, which enables covalent bonding with reactive groups of anionic dyes. Introducing cationic centers on the cellulose fiber or fabric enhances the migration of dyes toward the fiber/fabric. Interaction of the cationic polymer with cellulose prior to reactive dyeing brings several enhanced molecular features that contribute to improved color yields and dyeing operations compared to conventional methods: (a) Addition of salt can be reduced, or, in some cases, salt-free dyeing is possible. (b) Dyeing operations can be conducted at neutral pH. (c) Minimal to no addition of alkali discourages dye hydrolysis, one of the major challenges during conventional dyeing, where nearly 10–40% of dyes become hydrolyzed in alkaline conditions. (d) Reduced hydrolysis results in more dye availability in the dye bath, leading to higher exhaustion and improved color fixation, thus reducing colored wastewater effluent. (e) Higher fixation requires less washing. (f) Decrease in operational time. (g) Minimal wash-off led to a decrease in water usage volume.

Poly(vinylamine chloride) cationic polymer: Ma and colleagues investigated the salt-free dyeing of cationized cellulose material (cotton fabric) using poly(vinylamine chloride) (PVAC) with reactive dyes, as shown in Figure 10e [64]. The PVAC molecule contains numerous cationic sites with primary amino groups (H_3_N^+^), which are particularly effective for achieving salt-free dyeing. Increasing the pH proportionally decreases the abundance of H_3_N^+^ groups while proportionally increasing the availability of H_2_N groups. Cotton pretreatment with PVAC is carried out using a pad-bake method. Typically, 5 g/L of PVAC is applied to the cotton fabric, achieving an 80% wet pickup. The pretreatment liquor is buffered to a pH of 7.0 using potassium dihydrogen phosphate (7 g/L) and sodium hydroxide (1.39 g/L). The fabric samples are pre-dried at room temperature and then baked at 100 °C for 10 min in a rapid baker. The PVAC-treated cotton is then dyed with cold brand, hot brand, and vinyl sulfone reactive dyes using standard dyeing procedures. Compared to conventional dyeing with salt, PVAC-pretreated cotton exhibits salt-free dyeing with enhanced fixation with most of the reactive dyes. The operational accessibility of pretreatment of PVAC using the pad/air dry/bake method with dyeing protocols is simple and facilitates homogeneous dyeing. PVAC-pretreated cellulose materials (fabrics) exhibit a Langmuir adsorption, confirming its efficiency even at lower concentrations. The reactive dyes penetrate the fiber thoroughly, resulting in excellent washfastness and good crockfastness. Importantly, a significant reduction of colorant disposed of in effluent was observed, highlighting the potential of PVAC as a pretreatment chemical for cotton cationization in dyeing industries.

## 4. Natural or Biobased Cationizing Agents for Cellulose

### 4.1. Chitosan Derivatives as Cationizing Agents

There are various other cationized agents available for different fiber materials; however, a substantial improvement in the reactivity of natural fiber has been with synthetic or biased cationizing agents. Chitosan, being the second most common naturally available biopolymer after cellulose, can be integrated with cellulose-based materials and is primarily found in the exoskeletons of arthropods. It is produced by treating shrimp and other crustacean shells with sodium hydroxide. Its exceptional physicochemical properties, including biodegradability and biocompatibility, make it an ideal material for biomedical applications [65,66]. Structurally, chitosan is a linear chain containing amino polysaccharides, with randomly distributed β-(1→4)-linked D-glucosamine (deacetylated units) and N-acetyl-D-glucosamine (acetylated units), as shown in Figure 11 [65]. As a polycationic biopolymer, it exhibits a high surface charge density, enabling effective surface interactions with anionic polymers or molecules. In addition to its polycationic nature, chitosan’s structural features include polyhydroxy groups and extensive intermolecular hydrogen bonding. The presence of amino groups distinguishes it from cellulose-based materials, providing cationic properties and influencing its molecular arrangement and physicochemical characteristics, which are crucial for commercial processing. However, due to the high density of hydroxyl groups and resulting hydrogen bonding, chitosan lacks reasonable aqueous solubility. Despite this, its polycationic surface charge makes it effective in exhibiting antibacterial properties [65]. Cellulose-chitosan conjugates are known to exhibit an H-bond intermolecular interaction with each other in composite form but can also be covalently linked together through a cross-linker agent [67,68]. In this case, cyanuric chloride can be used as a cross-linker as chitosan has enough reactivity to form a covalent bond with cellulose materials. There has been substantial reporting of chitosan being a natural cationizing agent that is used to accommodate reactive dyeing as well as direct dyes as well. For example, a pad-dry-cure process chitosan nanoparticle-treated cotton has shown improvement in dyeability towards the acid dyes (where Acid Red 88 had a higher K/S value than Acid Blue 317) [13]. However, in my opinion, there must be a reason related to the solubility, as Acid Blue 317 reported with metal, which is coordinated with the organic ligands available in this dye. Therefore, the presence of positive charge metal might be responsible for ionic-ionic repulsion on the chitosan nanoparticle surface. Additionally, quaternary centers on the chitosan have intrinsic antibacterial and, also bring other activities such as it can work as a scaffold for delivery agents.

Cellulose materials can be exhausted with chitosan derivatives such as O-acrylamidomethyl-N-[(2-hydroxy-3-trimethylammonium)propyl] chitosan chloride (NMA-HTCC). Subsequently, NMA-HTCC-treated cellulose fibers (cotton) can undergo salt-free reactive or direct dyeing [69,70,71]. The resulting shades onto the NMA-HTCC-treated cellulose fibers exhibit better color strengths than the untreated cellulose fibers, even when a large amount of salt is added to the dyeing of untreated fibers. Post-dyeing, cellulose fibers treated with NMA-HTCC show improved wash fastness than untreated cellulose fibers, although its lightfastness is inferior. The antimicrobial activity of NMA-HTCC-treated cotton against *Staphylococcus aureus* is significantly reduced after dyeing, likely because the cationic group’s antimicrobial effect is blocked by its interaction with the anionic dye.

### 4.2. Pre-Treatment with Cationic Starch

Currently, nearly 15% of reactive dyeing at a global scale uses continuous dyeing methods. In this process, reactive dyes are directly padded onto cotton using a padding mangle. This eliminates the need for the high salt concentrations (ranging from 40–100 g/L) required to counteract dye-fiber repulsion in exhaust dyeing. However, a specified amount of salt (10–50 g/L) is still necessary to decrease dye migration, with higher salt concentrations being more effective [72]. Cationic starch is prepared using glycidyltrimethylammonium chloride (Glytac A), and its effect on cotton dyeing is examined using eight reactive dyes. Pretreating cotton with cationic starch enhances the dye fixation level in continuous dyeing. Better results are achieved at higher degrees of substitution, with the optimal amount of cationic starch ranging between 0.5% and 1.0% [73].

## 5. Other Methods or Materials of Cationized Cellulose Materials

### 5.1. Pigment Dyeing

Using pigments can achieve a weathered or distressed look for garments without the need for subsequent garment washing. Typically, the process involves dyeing the fabric in piece form, making up the garments, and then laundering them to wash down the color for the desired effect. This requires strict control over dye selection and application to ensure uniform and reproducible results. Alternatively, a garment-dyeing process has been developed to create garments with a weathered or distressed look without additional washing. Since pigments are water-insoluble and have limited affinity for cellulose-based fibers or materials, binders are required as chemical auxiliaries to fix these pigments onto the fiber surface, resulting in improved colorfastness. Reports have shown that using cationic agents to cationize cellulose-based fabrics or fibers (such as cotton) enhances the exhaustion of pigments. Most companies provide a comprehensive system of pigments or cationic agents that have been evaluated for their effectiveness and application conditions. It is crucial to select a system that ensures high pigment exhaustion and reproducibility. Other key factors to consider include color range, fastness, cost, and the desired outcome. Fabrics like twills, single-knit piques, or other textured surfaces are ideal for achieving the best results with the pigment-dyeing process. The recommended equipment for this process is typically a rotary drum machine that can be programmed with variable drum speeds during the dyeing cycle. However, other types of equipment, such as paddle machines, are also used in pigment garment dyeing.

First, the cationizing agent is applied to the garment, serving as a link between the fabric and the pigment. After the cationic treatment, the bath is drained, the garment is rinsed, and the pigment is applied. The liquor ratio (LR) is maintained at 20:1, and the initial bath temperature is kept low (27 °C) to ensure uniform pigment exhaustion onto the garment. A slow temperature increase (1 °C/min) further aids in achieving uniform pigment exhaustion. After rinsing off any excess color, a low-temperature or air-curable binder, typically an acrylic-based product, is applied to the garments to improve crockfastness. It is preferable to use a cationic binder rather than a nonionic one. Due to the potential difficulty in achieving reproducibility from one dye lot to the next, it is crucial to carefully select garment components, dyes, and chemicals and to closely monitor the dyeing process. Precise process control is essential to reproduce colors between dye lots within acceptable tolerances.

The exhaust method of pigment dyeing is commonly used for textiles made from both synthetic and natural fibers. Synthetic fibers include rayon (semi-synthetic), acrylic, polyester, and nylon, while natural fibers include cotton and protein fibers such as wool and silk. Natural fibers are primarily categorized into cellulose-based and protein-based types. Conventional pigment dyeing methods have several shortcomings, including poor wet crockfastness and a coarse handle. However, the exhaust method can achieve moderate dry crockfastness at medium color depths [42]. One strategy to mitigate issues in natural or pigment dyeing is to pretreat fabric materials by cationization. Introducing cationic centers enhances the interaction between pigments (which often have negatively charged groups like carboxylic acids or phenolic structures) and the fiber, resulting in better pigment adsorption on the fabric surface. Additionally, desizing the pigments can play a vital role in enhancing dyeability. By reducing the particle size, the surface area is significantly increased, leading to improved dispersion stability and better color yields. Pigment particles with reduced diameters (in the range of 100–200 nm) achieve superior results compared to dyeing with native-sized pigment particles [74]. Pigments are insoluble in water and are mixed with dispersing agents and cationic or anionic polymers for dyeing. This method is complex and often a trade secret for many companies. However, cationized pigment emulsions are less marketable due to their lower value-added benefits compared to anionized pigment emulsions. New types of pigments, known as pigment resin colors (PRC), have been developed for textile dyeing. These pigments are maintained in a stable dispersion in water using anionic surfactants. An anionic pigment solution usually holds three main ingredients [75]: (A) a pigment(s) with ranging 60–95%, (B) Anionic surfactant with ranging 5–40% that has aqueous solubility (could be copolymer or homopolymer), and (C) nonionic surfactant with ranging 0–20%. For example, a high-speed mixing of C.I. Pigment Red 22 and an anionic polymeric dispersant XG-1 resulted in a typical dispersion solution of pigment with an average particle diameter of approximately 150 Å. Later, using a microfluidizer, the nanoscale desizing of this dispersion solution resulted in an 11.7-fold reduction of its average particle diameter (12.8 Å). Cationization of cotton samples was achieved in a strongly alkaline medium of pH 11.0, with 30 min sustain temperature of 80 °C. Subsequently, to rinse the sample, cationic cotton is exhaust dyed with the prepared nano pigment dispersion at room temperature for 10 min, followed by the heating of the solution to 70–80 °C for 10 min. A binder is added after 20 min, and the process further sustains for an additional 10 min at the same temperature, followed by sample washing using nonionic detergent.

The study found that pigment uptake is significantly influenced by the amount added of the cationic reagent, the pH of the solution, the temperature of the pretreatment solution, and the time duration used for the pretreatment method. Moreover, these parameters are known to impact the colorfastness of pigment-dyed fabrics. Nano-dispersion solutions of pigments result in superior color properties compared to traditional dispersion solutions of pigments. The fastness profiles of such pigment dyeing are found to be adequate for wash-down effects. Exhaust dyeing with nanoscale pigments offers several benefits, such as a lesser amount required for operational pigment dyeing and improved handling and color yields, which ultimately provide ecological advantages [76]. Additionally, strategies involving pigment dyeing with nanoformulation or microformulation have shown improvements [77,78]. Reducing the size of pigment particles can enhance their exposed surface area, thereby increasing their dispersibility in aqueous solutions, also depending on the specific structural features of the pigment.

### 5.2. Cationized Linen Dyeing

Linen fabric materials, known as high cellulose, can achieve a facile cationization. In a study, the author used two quaternary ammonium cationic agents: CHPTAC (as discussed in the earlier part of the manuscript) and polyaminochlorohydrin quaternary ammonium salt with epoxide functionality (PAQAC). Cold pad batch dyeing with four types of reactive dyes onto the non-cationized and cationized linen fabric materials was performed. The cationization process with CHPTAC is pH-dependent and requires alkali. The fastness properties of the dyed fabrics are assessed, revealing that optimal K/S values are achieved with a 12-h batching time for both CHPTAC and PAQAC. Linen fabrics treated with CHPTAC exhibit deeper shades when dyed with the selected fiber-reactive dyes compared to those treated with PAQAC. The fastness properties (e.g., washing, rubbing, and perspiration) of the fabrics cationized with CHPTAC and PAQAC are nearly identical. The dyed fabrics treated with CHPTAC exhibit superior lightfastness compared to those treated with PAQAC, suggesting that CHPTAC cationization penetrates the fiber rather than remaining on the surface. In contrast, the poor lightfastness of fabrics treated with PAQAC indicates surface-level dyeing [79]. Two acid dyes and two direct dyes are used to dye linen fabrics, aided by cationization with two commercial agents: CHPTAC and polyhexamethylene biguanides (PHMB).

#### 5.2.1. Cold Pad Batch Method

Linen fabric is padded with dye, 250 g/L of CHPTAC, and 180 g/L of sodium hydroxide. The wrapped fabric is stored at room temperature for 12 h, then rinsed multiple times with water, followed by a rinse with a diluted acetic acid solution (1 g/L), and air-dried. The final pH of the fabric is 7.4.

#### 5.2.2. Pad Dry Cure Method

Linen fabric is padded with the solution to achieve a 95% pickup, then dried at 60 °C for 5 min. The dried fabric is cured at 160 °C for 2 min, rinsed several times with water, followed by a rinse with a diluted acetic acid solution (1 g/L), and air-dried. The final pH of the fabric is 7.4. Both unmodified and cationized linen fabrics are dyed with the selected dyes using three different dyeing methods. The fastness properties of the dyed fabrics are evaluated. Cationic treatment with CHPTAC and PHMB results in deeper shades when exhaust dyed with acid and direct dyes compared to unmodified linen fabric. Unmodified linen fabrics exhibit lighter shades when dyed under the same conditions [79].

### 5.3. Dyeing of Ramie Fiber

An application of in situ activation of CHPTAC using sodium hydroxide into its epoxide form that can attacked by the hydroxy groups of glucose units (cellulose part) of the ramie fibers studied. The post-treatment changes in morphology and structure of ramie fiber were studied using X-ray diffraction, differential scanning calorimetry, and thermogravimetric analysis. Although X-ray diffraction results indicate that the crystal structure of the modified fiber is still maintained after the post-processing using (CHPTAC + NaOH), a decrease in crystallinity was observed, as also supported by thermogravimetric analysis. How the (CHPTAC + NaOH) pretreatment affected the dyeing of ramie fibers was studied, where optimal conditions were noted as 30 g/L of CHPTAC, 15 g/L of NaOH, a reaction temperature of 50 °C, and a reaction time of 1 h. Consequently, pretreated (CHPTAC + NaOH) fibers and untreated were dyed using C.I. Reactive Red 2, where color yields of pretreated (CHPTAC + NaOH) fibers measured three times higher than those of the unmodified fiber. Interestingly, it has been noted that increasing the concentration of the CHPTAC enhances the dye uptake as nitrogen content (resulting from CHPTAC presence in the fiber) increases up to 0.4% on the pretreated fibers [80].

## 6. Potential Health Hazard Associated with Quaternary Salts

Most of these cationic agents possess the chemical functionality of quaternary ammonium, classifying them as quaternary ammonium compounds (QACs). QACs are well-known for their anti-infective and disinfectant properties and are commonly used as surfactants and preservatives. Generally, they are considered safe or cause only mild symptoms. However, individuals with prolonged exposure to these chemicals should take extra precautions, as long-term studies are still needed for more conclusive results. Being positively charged and often containing multiple hydrophobic organic groups, QACs are soluble in water but can also be present in a dispersed phase in polar protic solvents with high miscibility with water. This makes them easily dissolved in the environment. The increased use of QACs during the COVID-19 pandemic has highlighted some specific potential toxicities, such as allergies and skin irritation in humans. In non-human studies, they have shown disruptions in endocrine and immune functions and possess reproductive toxicity. Due to their stability and resistance to biodegradation, QACs persist in soil and surface/groundwater, posing ecological risks. There are reviews that compile more information about their potential toxicity, which readers may find useful [81,82,83].

## 7. Conclusions

The cationization of cellulosic fibers has garnered significant interest among researchers due to the need for eco-friendly processing solutions. Cellulose fibers are particularly important due to their widespread use in material-based applications. However, the industrial processing of cellulose materials produces a significant amount of contaminated wastewater. Consequently, recent technological advancements such as membrane-based filtration [84,85,86,87], photochemical oxidation [88,89], and biodegradation of dye water [90,91,92,93,94,95] are promising strategies to reduce water pollution from these industries. Although these technologies show an element of sustainable processing, they do not address the root cause of the wastewater issue. Therefore, researchers are keen to investigate new strategies that advance our scientific knowledge in dye-fiber chemistry, which could improve industrial processing and potentially minimize wastewater production. Cationization is a process that has garnered significant interest in cellulose processing in recent years. While most cationization processes have been developed for cellulose fibers, these methods can also be adapted for other cellulose applications, such as nanocellulose or nanocellulose fibrils, to enhance drug targeting, alter micelle solubility, or increase loading capacity. The coloration industry, which heavily relies on cellulose materials, faces challenges related to its ecological footprint. With growing awareness of the environmental impact of commercial products, the industry is seeking ways to reduce chemical waste. This has led to the development of strategies that can be implemented during pretreatment processes, such as using chemical auxiliaries other than dyes to improve the color properties of cellulose materials. Cationization is one such strategy, as it allows for deeper penetration into cellulose materials, particularly cellulose fibers. However, the dyeing of cationized fibers is not yet standardized and remains limited to a few small-scale dyers. Although the cationization of cotton is not fully commercialized, extensive research has been conducted in this area. It is now crucial to coordinate and integrate the various studies in this field.

## Figures and Tables

**Figure 1 polymers-17-00036-f001:**
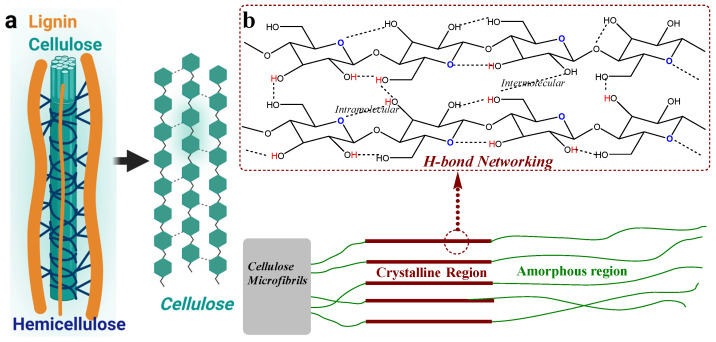
(**a**) Cellulose in natural form (coexisting with lignin and hemicellulose). (**b**) Network of intermolecular H-bonding within the cellulose structure.

**Figure 2 polymers-17-00036-f002:**
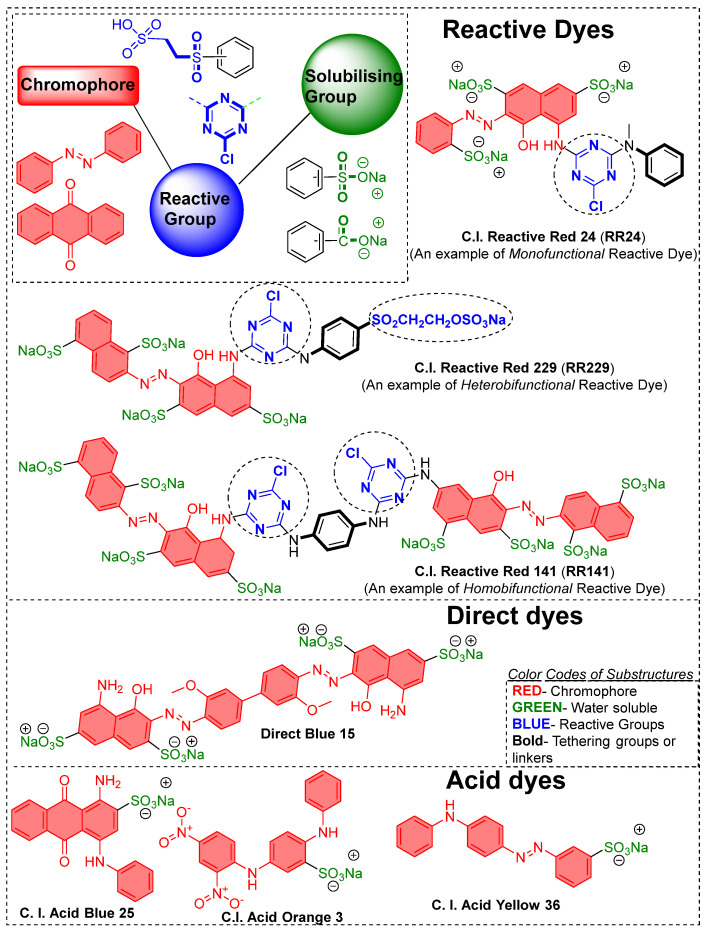
Structural relevance of anionic dyes: Anionic dyes can be categorized into three main types: reactive dyes (e.g., C.I. Reactive Red 24, C.I. Reactive Red 229, and C.I. Reactive Red 141), direct dyes (e.g., Direct Blue 15), and acid dyes (e.g., C.I. Acid Blue 25, C.I. Acid Orange 3, and C.I. Acid Yellow 36). These dyes primarily consist of chromophore and sulfate groups. Additionally, reactive dyes contain reactive groups in the form of monochlorotriazine and vinyl sulphone groups.

**Figure 3 polymers-17-00036-f003:**
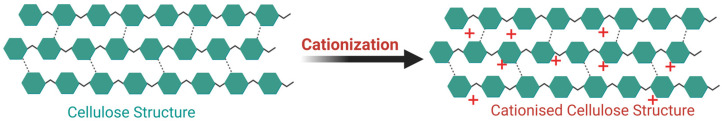
Schematic representation of cationized cellulose.

**Figure 4 polymers-17-00036-f004:**
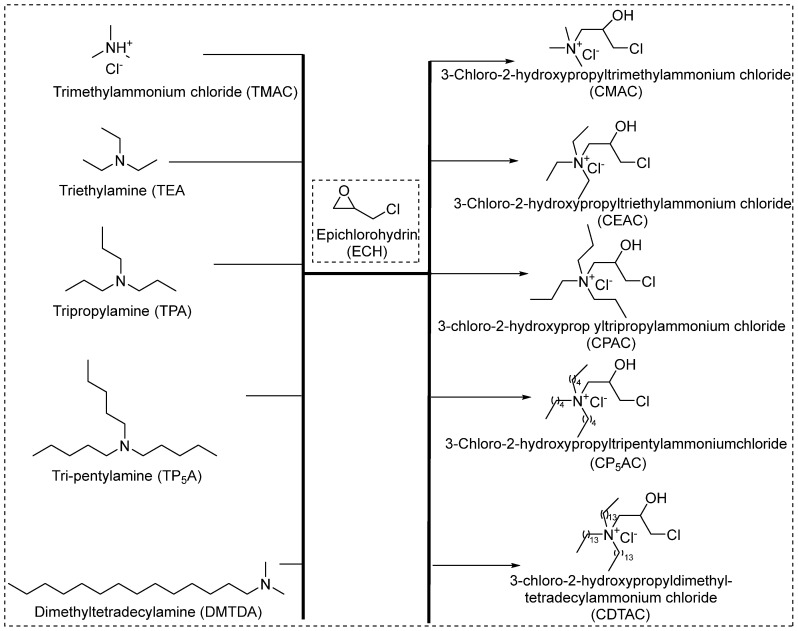
Reaction between various trialkyl amines with epichlorohydrin.

**Figure 5 polymers-17-00036-f005:**
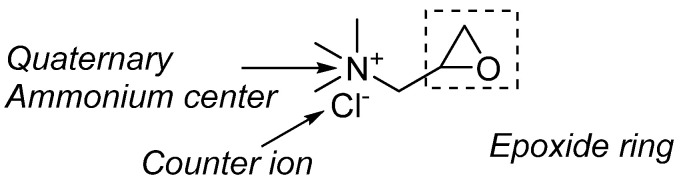
Molecular structure of EPTAC.

**Figure 6 polymers-17-00036-f006:**
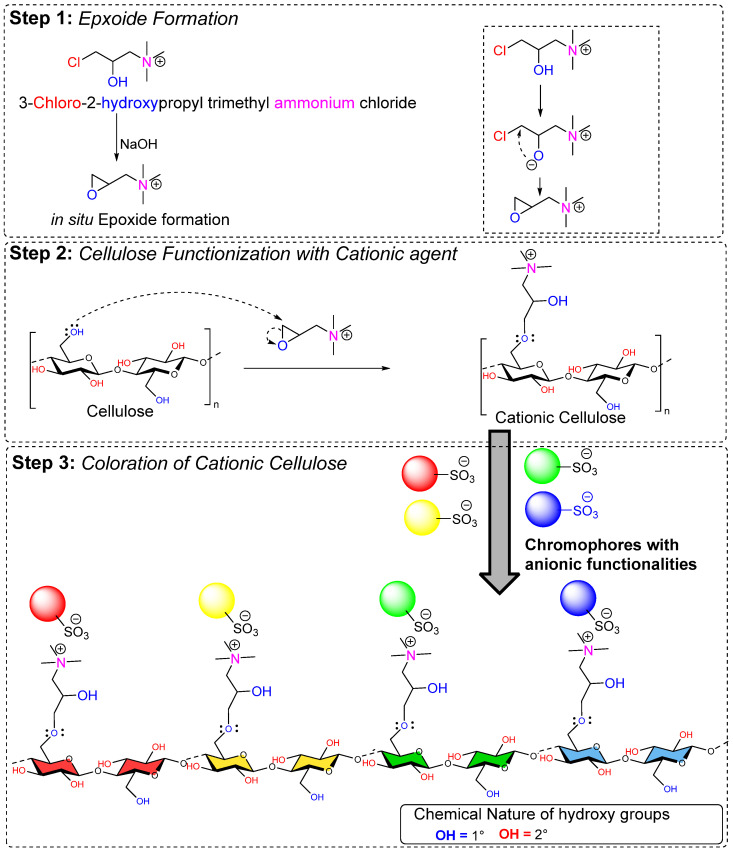
Reaction mechanism: The chlorohydrin form of the reagent (CHPTAC) is converted to the epoxy intermediate, EPTAC. In the second step, cellulose performs a nucleophilic attack on the EPTAC intermediate, opening the epoxide ring and exposing its cationic center, which becomes integrated into the cationized cellulose. The third step shows that these formed cationic centers over the cellulose material enhance its ionic interactions with various anionic dyes.

**Figure 7 polymers-17-00036-f007:**
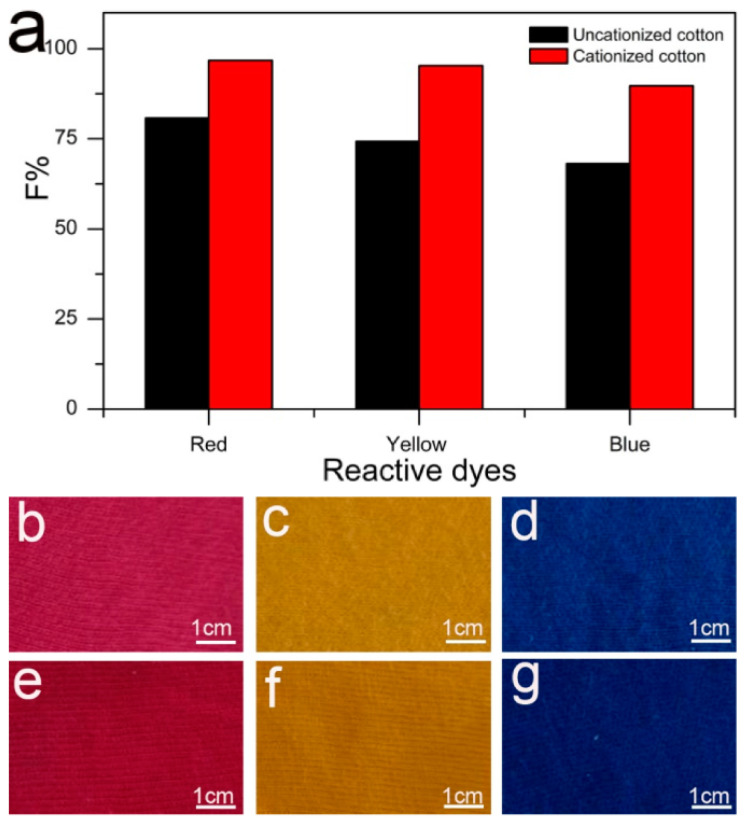
(**a**) Fixation rates of reactive dyes (C.I. Reactive Red 195, C. I. Reactive Yellow 145, and C. I. Reactive Blue 19) for cationized versus non-cationized samples. A comparison of color yields can be observed by comparing the samples (**b**) non-cationized or native cotton with C.I. Reactive Red 195; (**c**) non-cationized or native cotton with C. I. Reactive Yellow 145; (**d**) non-cationized or native cotton with C. I. Reactive Blue 19; (**e**) cationized cotton with C.I. Reactive Red 195; (**f**) cationized cotton with C. I. Reactive Yellow 145; (**g**) cationized cotton with C. I. Reactive Blue 19. The figure is reproduced from Ma et al. work [51].

**Figure 8 polymers-17-00036-f008:**
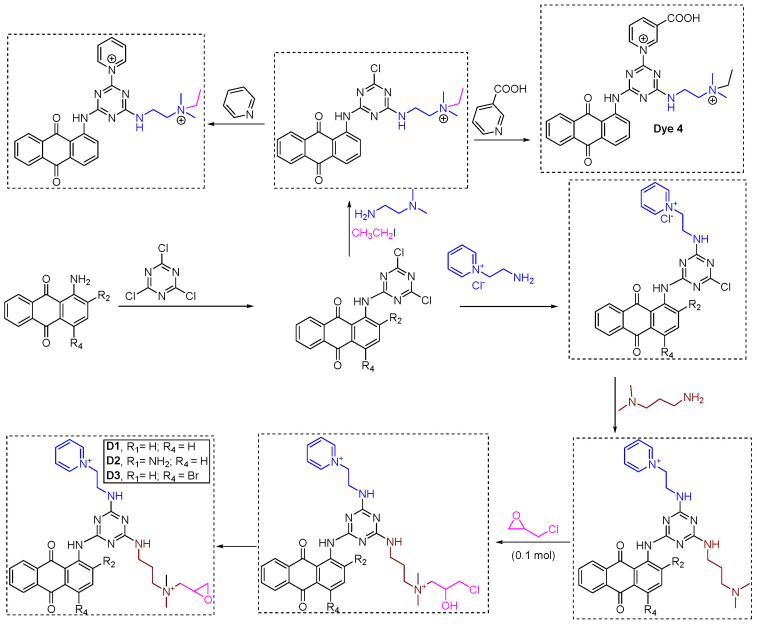
A multi-synthetic pathway of cationic reactive dyes.

**Figure 9 polymers-17-00036-f009:**
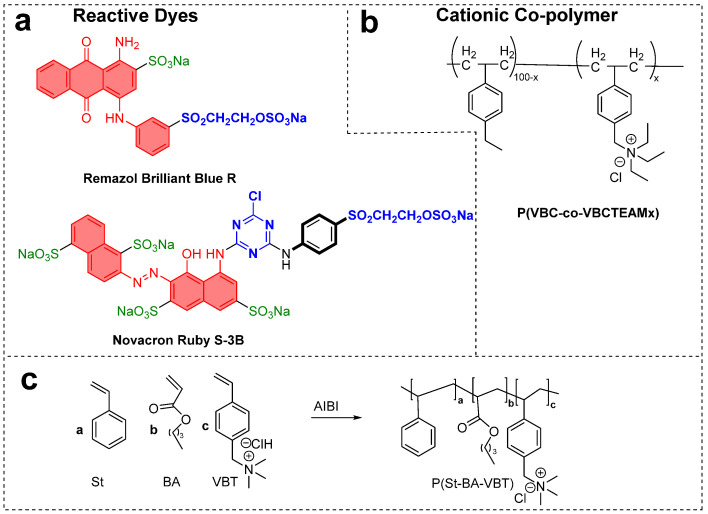
Cationic copolymers: (**a**) reactive dyes and (**b**) cationic polymers used by Papapetros et al. in their study [15]. (**c**) synthesis of cationic copolymer by Fang et al. [62].

**Figure 10 polymers-17-00036-f010:**
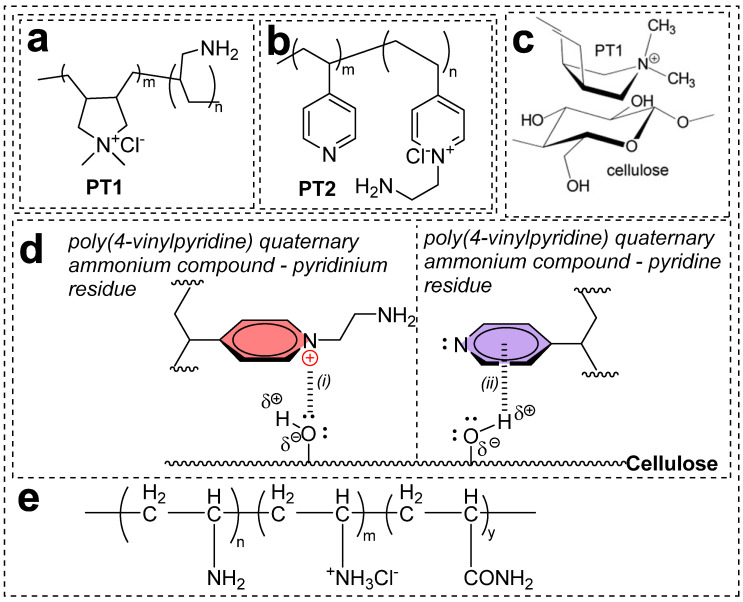
(**a**) Polymer (PT1) of diallyldimethyl ammonium chloride and aminopropene [63]. (**b**) Polymer (PT2) of quaternized vinylpyridine and aminochloroethane [63]. (**c**) Molecular interaction of cellulose structure (glucose unit) with polymer (PT1) [63]. (**d**) Ion–dipole interactions: one of the lone pair of oxygen atom hydroxy group of cellulose (could be C6, C2 or C3) interacting with pyridinium cationic center of polymer (PT2) using Ion-dipole interactions (i), while (ii) the hydrogen atom hydroxy group of cellulose (could be C6, C2 or C3) interacting (via Yoshida intermolecular Hydron bonding) with pyridine residues of polymer (PT2) [63]. (**e**) Molecular structure of polymer of vinylamine monomer units.

**Figure 11 polymers-17-00036-f011:**
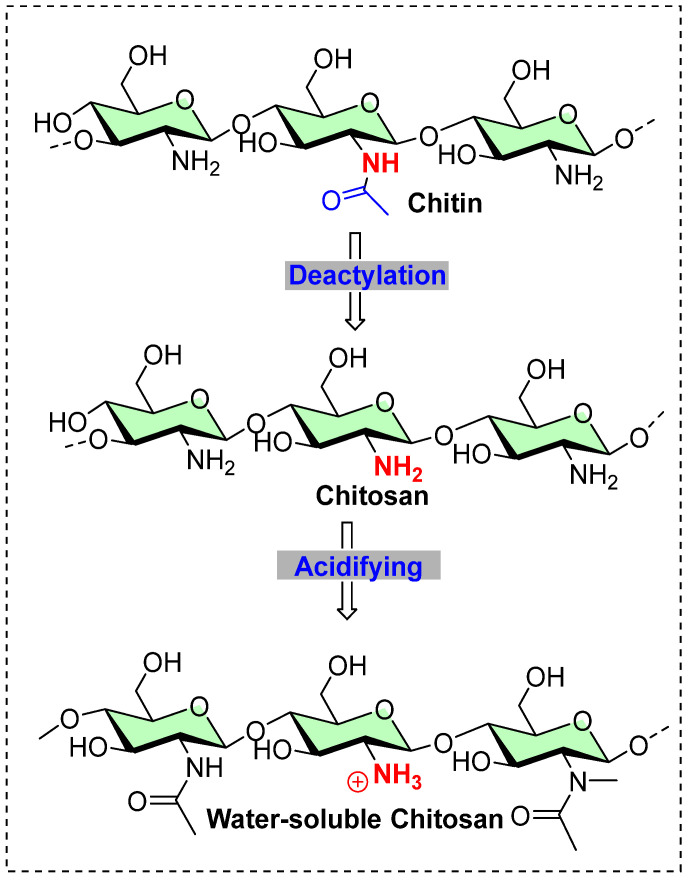
Chitosan is processed from its natural form (chitin) into a quaternary water-soluble form [65].

## Data Availability

Data sharing is not applicable.

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
