# Peer review of "Cationized Cellulose Materials: Enhancing Surface Adsorption Properties Towards Synthetic and Natural Dyes"

_polymers, 2024, doi:10.3390/polym17010036_

Round 1
Reviewer 1 Report
Comments and Suggestions for Authors
Reactivity of Cationised Cellulose Materials: Enhancing Surface Adsorption Properties towards Synthetic and Natural Dyes
This paper presents the review of Cationised Cellulose Materials production and interaction effectiveness with dyes. While the manuscript summarizes good and comprehensive work, several areas could be clarified to enhance readability.
1.. Please change title “Reactivity of Cationised Cellulose Materials …”. Reconsider term “cationised”.
2. Abstract is long, part of literature known data should be transferred to Introduction.
3. In subchapter “2. Reactivity of Industrial Cellulose Materials” author should add references.
4. Clarify “heterobifunctional” and “homobifunctional” and harmonize with Figure 2.
5. Be precise in “Reactive dyes, such as monochlorotriazine, dichlorotriazine, and vinyl sulphone, are commonly used in industrial applications” in term: reactive dyes containing moieties …”.
6. The same stand for “The high electrophilicity of reactive groups, due to the presence of electronegative elements like quaternary ions or halides, allows them to undergo nucleophilic attack reactions.” describe halides in the sense of the type of organohalides.
7. Schematic presentation in subchapter “3. Cationization of cellulose materials or fibers” is necessary.”
8. Please provide comparative pictures after dyeing to in order to distinguish the effectiveness of the processes. In the other hand, process given in Figure 9 is well known. In general, this manuscript suffers from additional figures and schemes. Advantages and drawbacks of mostly important/potential dyeing processes should be highlighted.
9. Minor English edition is necessary.
Author Response
Reactivity of Cationised Cellulose Materials: Enhancing Surface Adsorption Properties towards Synthetic and Natural Dyes
This paper presents the review of Cationised Cellulose Materials production and interaction effectiveness with dyes. While the manuscript summarizes good and comprehensive work, several areas could be clarified to enhance readability.
Author Response: The author would like to express gratitude to the reviewer for their thoughtful comments and for taking the time to provide feedback. To address the reviewer’s comments, enabled track changes as on, highlighting modifications in green for the reviewer’s or editor's convenience.
----------------------------------------List of comments ----------------------------------------------------
- Please change title “Reactivity of Cationised Cellulose Materials …”. Reconsider term “cationised”.
Author Response: As suggested the Reactivity word from the title has been removed, highlighted in green color.
- Abstract is long, part of literature known data should be transferred to Introduction.
Author Response: As suggested the abstract is rewritten and word count is reduced to 232 from 279, highlighted in green color.
“Abstract: Cellulose is a homopolymer composed of β-glucose units linked by 1,4-beta linkages in a linear arrangement, providing its structure with intermolecular H-bonding networking and crystallinity. The participation of hydroxy groups in the H-bonding network results in low-to-average nucleophilicity of cellulose, insufficient for executing nucleophilic reaction. Importantly, as a polyhydroxy biopolymer, cellulose has a high proportion of hydroxy groups in the form of secondary and primary, providing it with limited aqueous solubility, highly dependent on its form, size, and other materialistic properties. Therefore, cellulose materials are generally known for their low reactivity, and limited aqueous solubility, and usually undergo aqueous medium-assisted pretreatment methods. Cationization of cellulose materials is one such example of pretreatment, which introduces a positive charge over its surface, improving its accessibility towards anionic group-containing molecules or application-targeted functionalization. The chemistry of cationization of cellulose has been widely explored, leading to the development of various building blocks for different material-based applications. Specifically, in coloration applications, cationized cellulose materials have been extensively studied, as the dyeing process benefits from the enhanced ionic interactions with anionic groups (such as sulfate, carboxylic groups, or phenolic groups) minimizing/eliminating the need for chemical auxiliaries. This study provides insights into the chemistry of cellulose cationization, which can benefit the material, polymer, textile, and color chemist. This paper deals with the chemistry information of cationization and how it enhances the reactivity of cellulose fibers towards its processing. “
.
- In subchapter “2. Reactivity of Industrial Cellulose Materials” author should add references.
Author Response: As suggested, references are added to section 2.
- Clarify “heterobifunctional” and “homobifunctional” and harmonize with Figure 2.
Author Response: As suggested, the section has been rewritten to clarify the differences among the dyes. Accordingly, the figure 2 has also been modified.
- Be precise in “Reactive dyes, such as monochlorotriazine, dichlorotriazine, and vinyl sulphone, are commonly used in industrial applications” in term: reactive dyes containing moieties …”.
Author Response: The section has been rewritten with special attention given to the part about the reactive groups of reactive dyes. We thoroughly reviewed the manuscript, as various changes have been made throughout.
- The same stand for “The high electrophilicity of reactive groups, due to the presence of electronegative elements like quaternary ions or halides, allows them to undergo nucleophilic attack reactions.” describe halides in the sense of the type of organohalides.
Author Response: Previously reported information was inappropriate, as it must not be halide but a halogen substituent therefore, this sentence is rewritten and attempted to improve the sense for general audience “The high electrophilicity of reactive groups, due to the presence of electronegative elements like quaternary ions or halogen substituents attached to the carbon atom, allows them to undergo nucleophilic attack reactions. The propensity or facilitation of such nucleophilic attack on a specific highly electrophilic carbon atom is due to its attachment with a highly electronegative substituent (in form of atom, group or functionality). For example, the chemical structure of a monochlorotriazine group has a chloro-substituent on its electron-deficient triazine ring, provides a site for nucleophilic attack from cellulose, forming non-ionic covalent bonds."
- Schematic presentation in subchapter “3. Cationization of cellulose materials or fibers” is necessary.”
Author Response: A schematic presentation added as Figure 3 in the manuscript.
Figure 3. Schematic representation of cationised cellulose.
- Please provide comparative pictures after dyeing to in order to distinguish the effectiveness of the processes. In the other hand, process given in Figure 9 is well known. In general, this manuscript suffers from additional figures and schemes. Advantages and drawbacks of mostly important/potential dyeing processes should be highlighted.
Author Response: This comment has three parts
- For comparative pictures, we have added a new section with figures.
“Ma et al. from State Key Laboratory of Fine Chemicals, Dalian University of Technology, Dalian 116023, China [51] accomplish pretreatment and salt-free dyeing of greige knitted cotton fabrics using three reactive dyes (C.I. Reactive Red 195, C. I. Reactive Yellow 145, and C. I. Reactive Blue 19). To achieve cationization they used glycidyltrimethylammonium chloride, where Figure 7a showed fixation percentage of all the three reactive dyes when dyed to a cationised cotton sample (96.74% for C.I. Reactive Red 195, 95.29%, for C. I. Reactive Yellow 145, and 89.72% for C. I. Reactive Blue 19) as shown in Figure 7b-g.
Figure 7. (a) Fixation rates of reactive dyes (C.I. Reactive Red 195, C. I. Reactive Yellow 145, and C. I. Reactive Blue 19) for cationised versus non-cationised samples. A comparison of color yields can be observed by comparing the samples (b) non-cationised or native cotton with C.I. Reactive Red 195; (c) non-cationised or native cotton with C. I. Reactive Yellow 145; (d) non-cationised or native cotton with C. I. Reactive Blue 19; (e) cationised cotton with C.I. Reactive Red 195; (f) cationised cotton with C. I. Reactive Yellow 145; (g) cationised cotton with C. I. Reactive Blue 19. The figure is reproduced from Ma et al. work[51].
- As the authors, we kindly request to retain Figure 9 (now Figure 11) in the manuscript. Although the information is well-known, we have included it to provide a general overview for our audience before delving into the specifics of the topic. We would greatly appreciate it if the reviewers could consider keeping this figure, if possible.
(c) The sections detailing the dyeing methodology have now been updated to include information on the advantages and disadvantages.
- Minor English edition is necessary.
Author Response: The manuscript has been revised, and track changes have been enabled for the reviewers' convenience

Reviewer 2 Report
Comments and Suggestions for Authors
Summary: In this review paper, the author covers a broad discussion on the reported cationization methods for cellulose materials and their effects on surface adsorption properties, particularly for reactive and direct dyes. The topic of this review paper fits into the scope of the Polymers journal. However, the reviewer thinks that the manuscript requires a minor revision before it can be considered for publication.
Comments:
1. Section 1. Introduction, section 2. Reactivity of Industrial Cellulose Materials.
There are several places in those two sections lacking relevant references to support the statement made. For example, in line 56 to 59, where the applications of cationized cellulose in antimicrobial fishing and others are mentioned, the proper reference should be cited here. The reviewer recommends that the author review those sections and include relevant references in the right places.
2. Figure 1. The figure lacks clear annotation for part (a) and (b). Please ensure that these annotations are added.
3. Figure 2. The legend of color codes seems redundant as the same information is already presented in the top-left part of the figure. Please consider removing it.
4. line 131 to 134. The first sentence starting with “Being anionic in nature…” seems to provide overlapping information with the subsequent sentence starting with “Additionally, xxx”. Please rewrite those two sentences and avoid redundancy.
5. Figure 7c and 8d. Those two figures appear to be blurred. Please use the high-resolution figures.
Author Response
Summary: In this review paper, the author covers a broad discussion on the reported cationization methods for cellulose materials and their effects on surface adsorption properties, particularly for reactive and direct dyes. The topic of this review paper fits into the scope of the Polymers journal. However, the reviewer thinks that the manuscript requires a minor revision before it can be considered for publication.
Author Response: The author would like to express gratitude to the reviewer for their thoughtful comments and for taking the time to provide feedback. To address the reviewer’s comments, we have included continuous line numbering and enabled track changes, highlighting modifications in green for the reviewer’s or editor's convenience.
----------------------------------------Comments ----------------------------------------------------
- Section 1. Introduction, section 2. Reactivity of Industrial Cellulose Materials.
There are several places in those two sections lacking relevant references to support the statement made. For example, in line 56 to 59, where the applications of cationized cellulose in antimicrobial fishing and others are mentioned, the proper reference should be cited here. The reviewer recommends that the author review those sections and include relevant references in the right places.
Author Response: References are added to those sections. Please follow the track changes version of manuscript.
- Figure 1. The figure lacks clear annotation for part (a) and (b). Please ensure that these annotations are added.
Author Response: The annotation are now added to the figure 1.
- Figure 2. The legend of color codes seems redundant as the same information is already presented in the top-left part of the figure. Please consider removing it.
Author Response: The authors would like to express their appreciation for the reviewer's keen observation regarding the color coding of different parts of the dyes. While the color codes are clear, certain parts that are not commonly present in dye structures (such as tethering parts) could potentially cause confusion among readers. To simplify this, we have provided legends within the figure to help readers easily navigate the different parts of dye structure. Please refer to the changes made in Figure 2.
- line 131 to 134. The first sentence starting with “Being anionic in nature…” seems to provide overlapping information with the subsequent sentence starting with “Additionally, xxx”. Please rewrite those two sentences and avoid redundancy.
Author Response: The sentence is rephrased to remove redundancy, highlighted in green color.
- Figure 7c and 8d. Those two figures appear to be blurred. Please use the high-resolution figures.
Author Response: These subsections of figures (7c and 8d) are made in chemdraw and the numbering is changed to 9c and 10d.
